# Combined Effects of Lycopene and Metformin on Decreasing Oxidative Stress by Triggering Endogenous Antioxidant Defenses in Diet-Induced Obese Mice

**DOI:** 10.3390/molecules27238503

**Published:** 2022-12-03

**Authors:** Bruno Pereira Motta, Camila Graça Pinheiro, Ingrid Delbone Figueiredo, Felipe Nunes Cardoso, Juliana Oriel Oliveira, Rachel Temperani Amaral Machado, Patrícia Bento da Silva, Marlus Chorilli, Iguatemy Lourenço Brunetti, Amanda Martins Baviera

**Affiliations:** 1Department of Clinical Analysis, School of Pharmaceutical Sciences, São Paulo State University (UNESP), Araraquara 14800-903, São Paulo, Brazil; 2Department of Drugs and Medicines, School of Pharmaceutical Sciences, São Paulo State University (UNESP), Araraquara 14800-903, São Paulo, Brazil

**Keywords:** paraoxonase-1, carotenoids, advanced glycation end products, glycoxidative stress, metabolic memory

## Abstract

Since lycopene has antioxidant activity, its combination with metformin may be useful to contrast diabetic complications related to oxidative stress. This study aimed to investigate the effects of metformin combined with lycopene on high-fat diet (HFD)-induced obese mice. Seventy-two C57BL-6J mice were divided into six groups: C (control diet-fed mice), H (HFD-fed mice for 17 weeks), H-V (HFD-fed mice treated with vehicle), H-M (HFD-fed mice treated with 50 mg/kg metformin), H-L (HFD-fed mice treated with 45 mg/kg lycopene), and H-ML (HFD-fed mice treated with 50 mg/kg metformin + 45 mg/kg lycopene). Treatments were administered for 8 weeks. Glucose tolerance, insulin sensitivity, fluorescent AGEs (advanced glycation end products), TBARS (thiobarbituric acid-reactive substances), and activities of antioxidant enzymes paraoxonase-1 (PON-1; plasma), superoxide dismutase, catalase and glutathione peroxidase (liver and kidneys) were determined. Metformin plus lycopene reduced body weight; improved insulin sensitivity and glucose tolerance; and decreased AGEs and TBARS in plasma, liver and kidneys. Combined therapy significantly increased the activities of antioxidant enzymes, mainly PON-1. Lycopene combined with metformin improved insulin resistance and glucose tolerance, and caused further increases in endogenous antioxidant defenses, arising as a promising therapeutic strategy for combating diabetic complications resulting from glycoxidative stress.

## 1. Introduction

Oxidative stress has a central role in the development of comorbidities and complications of obesity, including insulin resistance and type 2 diabetes mellitus (T2DM) [1,2]. The prevalence of T2DM is rising globally. In 2021, diabetes affected approximately 537 million adult people; by 2045, this number is predicted to rise to 783 million. It must be highlighted that T2DM accounts for over 90% of diabetes worldwide [3]. 

Hyperglycemia, dyslipidemia, chronic inflammation, and mitochondrial dysfunction are main contributors for developing oxidative stress in obesity [4] and T2DM [5]. In the milieus of the metabolic disturbances observed in obesity and diabetes, the exacerbated formation of advanced glycation end products (AGEs) contributes to the onset of oxidative stress [6]. Furthermore, during obesity and diabetes, oxidative stress also induces glycoxidation reactions that further increases AGE formation [7], becoming a vicious cycle.

In addition to an early and intensive treatment of hyperglycemia, the introduction of therapeutic agents that are able to reduce glycoxidative stress may be useful to contrast the events related to the “metabolic memory”, which represents the appearance of long-term complications in patients having metabolic diseases even when the glycemia is controlled successfully [8]. Metformin is the preferred initial pharmacological agent for T2DM treatment [9]; it is generally well tolerated and has recognized efficiency in the glycemic control. Nevertheless, about 35% patients with T2DM fail to reach glycemic goals with standard doses using metformin as monotherapy, which must require increased doses that may favor the risk of side effects [10]. Furthermore, the progressive nature of T2DM leads to deterioration of glycemic control and generally requires a gradual intensification of therapy in order to maintain adequate glycemic targets [11]. It has been proposed that an effective T2DM treatment requires multiple strategies, including combination of drugs to correct multiple pathophysiological impairments, in addition the treatment must be started early in prediabetic or diabetic patients to prevent progressive pancreatic β-cell failure [12]. These aforementioned considerations motivated the present study, which proposes a combined therapy between metformin and a natural bioactive in a scenario that precedes T2DM.

Lycopene (C_40_H_56_) is a carotenoid found in red fruits and vegetables, particularly tomatoes. The long carbon chain with 11 conjugated double bonds confers to lycopene a potent ability to scavenge reactive oxygen species (ROS) [13]. Lycopene mitigates oxidative damage and increases antioxidant status in T2DM and obesity, an effect not only associated to its per se antioxidant potential, but also to its ability to induce endogenous antioxidant defenses, mainly by enhancing antioxidant enzymes [14,15]. Furthermore, studies demonstrated that lycopene provided protection against metabolic complications resulting from advanced glycation [16,17].

There is well-established data on the effectiveness of combined therapies between antidiabetic drugs as a strategy for achieving the glycemic control, with main advantages the delay in the disease progression, the preservation of pancreatic β-cell function, and the attenuation of the deleterious impacts of metabolic memory, thus reducing the risk of diabetic complications [18]. However, more research is needed to advance our knowledge on the beneficial effects of combining natural bioactives with antihyperglycemic drugs in view of the extensive evidence in the literature about the properties of nutraceuticals and medicinal plants to prevent complications observed in chronic metabolic diseases, such as T2DM [19,20]. Thus, the present study aims to evaluate the effects of metformin combined with lycopene on biomarkers related to metabolic disturbances, glycoxidative stress, and antioxidant defenses in mice with induced obesity and insulin resistance. 

## 2. Results

### 2.1. Lycopene Combined with Metformin Reduced Body Weight Gain and Fat Deposition in Obese Mice 

At the beginning of the experiment (week 0), all mice had similar mean body weight values (Table 1). After 9 weeks of feeding a high-fat diet (HFD), and before the beginning of the treatments, all mice had a significant body weight gain (around 7.7–9.2 g) compared to the weight gain of mice fed with the control diet (C) (around 5.1 g), which characterizes the establishment of the in vivo model system of obesity. After the beginning of the treatments and until the end of the 17th week, mice fed the C diet (C group), or mice fed the HFD and not treated (H group) had a decrease in the body growth rate (Table 1). From the 9th to the 17th weeks, C mice practically stopped growing, while H mice had a body weight gain of 2.7 g. It is common to observe a decrease in the body growth rate of control animals, which can be attributed to the stress caused by the daily gavage (manipulation performed to mimic the treatments of the other groups). Overall, after 17 weeks, mice from H group underwent increases in body weight (Figure 1A), mostly from the 4th week. After 17 weeks, H mice had a mean body weight of 33.88 g, while mice from C group had a mean body weight of 26.74 g (Table 1). Thus, when compared to C mice, H mice had 2.5-fold increase in total body weight gain (Figure 1B). Increases were observed in the weights of epididymal (eWAT) and retroperitoneal (rWAT) white adipose tissues, liver, and kidney (Table 2). Weights of interscapular brown adipose tissue (BAT), heart and gastrocnemius muscle were not different among groups (Table 2). Although H mice exhibited decreased food intake (Figure 1C), they had increased energy intake (Figure 1D).

HFD-fed mice receiving vehicle (H-V group) had virtually the same changes as those of H group, with exception of a slight decrease (not statistically significant) in the body weight gain from the 14th to the 17th weeks of experiment (Figure 1A), and the decreases in the weights of eWAT and rWAT (Table 2); however, these effects were much smaller than those of metformin and/or lycopene.

Before the beginning of the treatments (i.e., until week 9), mice that would later be part of the groups treated with metformin and/or lycopene had a body growth rate similar to that observed in mice from the H group (Table 1), which indicates that all animals receiving HFD in the first 9 weeks were obese. However, after the beginning of the treatments and until the 17th week, HFD-fed mice treated with metformin (H-M group) or with lycopene (H-L group) had body weight loss (between 1.1 and 2.6 g) (Table 1). This body weight loss of animals after the beginning of the treatments caused decreases in the terminal body weight values (Figure 1A), leading to 52% (H-M) and 59% (H-L) decreases in total body weight gain when compared to H mice (Figure 1B). The weights of eWAT, rWAT, liver, and kidney were also decreased in H-M and H-L mice (Table 2). The low body weight gain of H-M or H-L groups may be a consequence, at least in part, of the slight decreases in the food intake from the 3rd week of treatment (12th week of experiment) (Figure 1C). However, the decreases in the food intake of groups treated with metformin and/or lycopene did not imply decreases in the mean energy intake monitored during the treatment period (Figure 1D), which were similar to the corresponding values of H group. 

Mice treated with metformin + lycopene (H-ML group) had body weight loss from the 9th to the 17th weeks of the experiment (Table 1), in the same magnitude as the effects of the isolated treatments. Furthermore, by combining metformin with lycopene, the beneficial effects of metformin or lycopene alone on body weight gain (Figure 1A), food intake (Figure 1C) and weights of tissues (Table 2) were maintained, without synergistic effects.

### 2.2. Lycopene Combined with Metformin Improved Insulin Sensitivity and Glucose Tolerance in Obese Mice

H mice developed insulin resistance and glucose intolerance. In the oral glucose tolerance test (OGTT), 15 min after the glucose challenge, H mice exhibited a higher hyperglycemic peak (36%) than that of C mice (Figure 2A). After 120 min, H mice did not correct the glycemia (124 ± 5.3 mg/dL) with the same efficiency as the C group (78 ± 4.4 mg/dL) (Figure 2A). In the insulin tolerance test (ITT), after insulin administration, high glycemia levels throughout the monitoring period were observed in H mice (Figure 2C), suggesting decreased insulin sensitivity (Figure 2D) that corroborates the glucose intolerance (Figure 2B). Mice from H-V group had the same impairments that those observed in H mice (Figure 2).

The treatments of HFD-fed mice with metformin or lycopene, individually, were effective to improve both the insulin sensitivity (Figure 2C,D) and the glucose tolerance (Figure 2A,B), although they did not reach the profile of the C group.

The glucose tolerance and the insulin sensitivity of HFD-fed mice treated with metformin + lycopene were improved in comparison to H group, as the combined therapy was as efficiently as the treatments with metformin or lycopene alone, suggesting maintenance of the beneficial effects of the isolated therapies (Figure 2).

### 2.3. Lycopene Combined with Metformin Reduced Glycoxidative Stress in Plasma and Significantly Increased the Activity of Paraoxonase-1 (PON-1) in Obese Mice

H mice had increased plasma levels of total-cholesterol (40%) and high-density lipoprotein-cholesterol (HDL-cholesterol; 37%), however the plasma levels of triglycerides were 21% decreased when compared to C group (Table 3). There is also evidence of a hepatic injury, with increased plasma levels of alanine aminotransferase (ALT; 50%) and alkaline phosphatase (ALP; 31%), while the levels of creatinine and albumin were unchanged. The levels of glycoxidative stress markers in H mice increased as well: the plasma levels of fluorescent AGEs (Figure 3A) and thiobarbituric acid reactive substances (TBARS; Figure 3B) increased by 34% and 45%, respectively, in H mice compared to those in C mice. Additionally, the activity of PON-1 in H mice reduced by 25% (Figure 3C). H-V mice had the same impairments that those observed in H mice relative to biochemical parameters and glycoxidative stress biomarkers in plasma.

The treatments with metformin or lycopene, individually, promoted various improvements in plasma biochemical parameters of HFD-fed mice. Both treatments decreased plasma levels of cholesterol and hepatic injury markers to a similar extent (Table 3). Levels of AGEs (Figure 3A) and TBARS (Figure 3B) were significantly decreased and the PON-1 activity was increased (Figure 3C), suggesting that these treatments were effective in decreasing glycoxidative stress. It is important to note that treatment with lycopene alone caused a significant increase in the PON-1 activity, which reached values even higher than those of mice from C group (Figure 3C).

The effects of the combined therapy on plasma biochemical markers were interesting: the treatment with metformin + lycopene reduced the plasma levels of cholesterol, AST, ALT (Table 3), and biomarkers of glycoxidative damage (Figure 3A,B) in a similar extend of the isolated treatments. On the other hand, the benefits of the combined therapy in the PON-1 activity were better than those of the isolated treatments, causing a very impressive increase in its activity, suggesting a synergistic effect (Figure 3C).

### 2.4. Lycopene Combined with Metformin Triggered Endogenous Antioxidant Defenses in Liver and Kidneys of Obese Mice

H mice exhibited increased levels of fluorescent AGEs in liver (33%; Figure 4A) and kidney (24%; Figure 5A) and TBARS in liver (56%; Figure 4B) and kidney (56%; Figure 5B). H mice also had significant decreases in the activities of the antioxidant enzymes in liver and kidneys, including superoxide dismutase (SOD; Figure 4C and Figure 5C), catalase (CAT; Figure 4D and Figure 5D) and glutathione peroxidase (GSH-Px; Figure 4E and Figure 5E). In sum, these changes support the onset of oxidative stress in the liver and kidneys of H mice. Mice from the H-V group had virtually the same changes as those of H group.

The treatment of HFD-fed mice with only metformin reduced the hepatic levels of AGEs (20%; Figure 4A) and TBARS (26%; Figure 4B), respectively, but failed to increase the activities of the antioxidant enzymes in this tissue (Figure 4C–E). In the kidneys, the decreases in the levels of AGEs (Figure 5A) and TBARS (Figure 5B) in metformin-treated mice were accompanied by an increase in the GSH-Px activity (Figure 5E), while the activities of SOD (Figure 5C) and CAT (Figure 5D) remained low.

The treatment with lycopene alone increased the activities of all antioxidant enzymes in liver (Figure 4C–E) and kidneys (Figure 5C–E) of HFD-fed mice. These beneficial effects on antioxidant enzymes were accompanied by decreases in the levels of AGEs (Figure 4A and Figure 5A) and TBARS (Figure 4B and Figure 5B) in the liver and kidneys of H-L mice.

The combined therapy added benefits to metformin regarding the capacity to contrast oxidative stress: the inability of metformin alone to prevent losses in the activities of antioxidant enzymes in liver (SOD, CAT and GSH-Px) and kidneys (SOD and CAT) of obese mice was overcome by the association with lycopene. All the lycopene beneficial effects on stimulating the antioxidant enzymes in liver (Figure 4C–E) and kidneys (Figure 5C–E) of HFD-fed mice were preserved in the combined therapy, reaching values even higher than those of C mice, as in the case of SOD activity in kidneys. Consequently, low levels of AGEs (Figure 4A and Figure 5A) and TBARS (Figure 4B and Figure 5B) were observed in liver and kidneys of mice treated with metformin + lycopene.

## 3. Discussion

The present study provides insights regarding the effects of the combined therapy between metformin and lycopene on disturbances occurring during obesity and insulin resistance, with emphasis on oxidative stress. Some beneficial effects of this therapy were achieved via maintenance of the best effects of the isolated treatments, including improvements in the glucose tolerance and insulin sensitivity, and the reduction in biomarkers of glycoxidative stress. This can be interpreted as a positive result, since the effects of these bioactives were not nullified when administered in combination. Furthermore, the main novelty of this study was the observation of the powerful in vivo antioxidant effects of lycopene when administered in combination with metformin; lycopene triggered cytoprotective machinery involved in the detoxification of ROS, which go beyond its well-known per se antioxidant capacity. In this regard, when administered to obese mice in combination with metformin, lycopene significantly increased endogenous antioxidant defenses, mainly the activities of PON-1 (plasma), SOD, CAT, and GSH-Px (liver and kidneys). It is important to mention that some synergistic effects were achieved when combining lycopene and metformin, highlighting the marked increase in the PON-1 activity. 

A previous study from our laboratory showed the benefits of combining lycopene and metformin in streptozotocin-induced diabetic rats [21], an experimental model that mimics features of T1DM. Various benefits of the combined therapy were observed, compared to metformin alone, including better management of glycemia and dyslipidemia, decrease in glycoxidative stress biomarkers and increase in endogenous antioxidant defenses. However, in this previous study [21], various impairments in diabetic rats were not completely prevented by lycopene + metformin, since the effects of the combined therapy did not reach the therapeutic efficacy of 4U/day insulin, considered in this study the gold standard therapy; this can be attributed to the severity of the experimental model. Therefore, the present study shows the benefits of metformin + lycopene in obesity and insulin resistance, a scenario that precedes T2DM. According to our findings, metformin + lycopene restored to normality levels (with reference to the C group) many of the parameters changed in obesity (with reference to the H group), which denotes the effectiveness of this combined therapy, when early introduced, to prevent complications of metabolic dysfunctions. Surely, care must be taken to extrapolate these findings to the obesity in humans, which is much more complex than the experimental model used, which is performed under strictly controlled conditions. 

By regulating various signaling pathways and having antioxidant and anti-inflammatory properties, lycopene is able to reduce the risk of obesity and diabetes and their associated complications [22,23]. In agreement with this, our findings showed that lycopene + metformin significantly improved the glucose tolerance and insulin sensitivity in HFD-fed mice; such effects were virtually the same as those of metformin or lycopene alone. Although not explored in the present study, there is evidence that lycopene exerts protective effects on pancreatic β-cells [24], which can be important to reduce the pancreatic dysfunction that precedes the establishment of T2DM. In addition, knowing that weight loss is a powerful means to improve glucose homeostasis in obesity and T2DM, the antiobesogenic effects of metformin and lycopene may also explain the improvements observed in the glucose tolerance of HFD-fed mice treated with the combined therapy. Our findings indicate that, from the moment treatments with metformin and/or lycopene were started, obese mice lost weight, which reinforces the antiobesogenic effects of these bioactives. Furthermore, the antiobesogenic effects of metformin and lycopene have similar profiles among isolated and combined therapies. The literature already reports the antiobesogenic effects of metformin [25] and lycopene [26]. According to our data, the reductions in the body weight gain and fat deposits observed in mice treated with metformin and/or lycopene can be attributed, at least partially, to the slight decreases observed in the food intake from the third week of treatment, although the mean energy intake monitored during the treatment period did not decrease in these animals. Previous studies reported that both metformin [25] and lycopene [27] increase energy expenditure, which may also be useful to explain our findings about the reductions in the body weight.

It must be highlighted the lycopene effects on stimulating the activities of antioxidant enzymes in the liver and kidneys of obese mice, which were preserved in the combination with metformin, adding antioxidant properties to the combined therapy. It is plausible that the antihyperglycemic effect of lycopene contributed to these effects, by decreasing the glycation of antioxidant enzymes; it has been observed that glycation processes inactivate antioxidant enzymes [28]. Previously, Tabrez et al. [16] observed that the treatment with lycopene decreased the AGE formation in kidneys of rats receiving ribose. The authors mentioned that this effect might not be attributed to an anti-AGE activity of lycopene, but due to its ability to scavenge ROS that led to the suppression of AGE production, based on the knowledge that AGEs and ROS are formed interchangeably in a vicious cycle. In addition to decreasing AGE levels, there is also evidence that lycopene decreases the expression of the RAGE receptor, thereby decreasing the AGE-RAGE axis activation and its deleterious effects [16,17]. AGE-RAGE axis is reported to increase the expression and activity of NADPH oxidase, leading to increased generation of ROS and depletion of cellular antioxidant defenses [29,30]. Thus, by decreasing RAGE expression, lycopene may contribute to increase the activities of antioxidant enzymes. Finally, it is also possible that lycopene stimulated the gene expression of these enzymes. Studies demonstrated that lycopene potentially protected tissue injuries caused by oxidative stress by enhancing the expression of antioxidant enzymes via activation of Nrf2 [31,32]. Nrf2 (nuclear factor (erythroid-derived 2)-like-2 factor) is a transcription factor involved in the regulation of the intracellular redox homeostasis that induces the expression of antioxidant and cytoprotective enzymes under conditions of oxidative stress.

The effects of the combined therapy on PON-1 activity are worth mentioning, since the PON-1 activity in obese mice treated with metformin + lycopene was significantly increased, reaching values even higher than those of mice fed a non-obesogenic diet. Furthermore, considering that metformin and lycopene alone also increased PON-1 activity, it is likely that the combined therapy had synergistic effects. In a recent review by Otocka-Kmiecik [33], there is evidence for the ability of lycopene and other carotenoids to increase PON-1 activity via upregulation of gene expression. Apart from increasing PON-1 activity, the combined therapy had favorable effects on decreasing cholesterol levels. Since PON-1 has been associated with atheroprotective effects [34], by decreasing cholesterol levels and increasing PON-1 activity metformin + lycopene may have protective effects on cardiovascular system.

Finally, an interesting approach of our study was the possibility of orally administering the combination of metformin + lycopene in a nanostructured lipid system. The use of nanostructured lipid system circumvents limitations related to the poor lycopene stability, since the carotenoid can be isomerized or degraded when exposed to light, oxygen and temperature variations. In addition, lycopene has low water solubility and poor oral bioavailability [35]. Another advantage to use the nanostructured lipid system is the possibility to have a stable pharmaceutical formulation containing two bioactive compounds (lycopene and metformin) with distinct water solubility. Furthermore, it is recognized that metformin formulations with delayed release are helpful to improve the glucose lowering action of metformin by favoring the drug effects on the gut [36], which may be a mechanism of the nanostructured lipid system used in this study. Notably, various nanotechnological preparations have been proposed to increase lycopene bioavailability and its biological effects [37,38]. However, as far as we know, this is the first study to investigate the beneficial health effects of lycopene administered into a nanostructured lipid system in combination with metformin as a strategy to improve the pharmacological effects of these bioactives.

The large body of clinical evidence about the lycopene benefits in terms of glycemic control and the reduction in oxidative stress biomarkers in patients with diabetes and obesity [22,39] reinforces the therapeutic potential of the combination between metformin and lycopene to contrast the diabetic complications. Recent review [15] brings several clinical studies about the lycopene effects on the glycemic control, biomarkers of oxidative damage and antioxidant status in T2DM patients under supplementation with lycopene or with lycopene-rich foods. Although the findings of these studies were not completely consistent in regard to the lycopene effects on improve glycemia in patients with T2DM (which can be attributed to several factors, including disease severity, food sources of the carotenoid, and supplementation period), the review reports that supplementation with lycopene was sufficient to attenuate oxidative damage by increasing antioxidant enzymes and decreasing lipid peroxidation rate in diabetic subjects.

## 4. Materials and Methods

### 4.1. Nanostructured Lipid System Preparation

The nanostructured lipid system was prepared according to de Freitas et al. [40] with modifications with the following composition: sunflower oil (5%) as the oil phase, surfactant mixture Brij O20/soy phosphatidylcholine–2:1 (10%), and phosphate buffer (pH 7.4) as the aqueous phase (85%). Lycopene and metformin were incorporated, alone or in combination, to the oil phase and surfactant mixture. The aqueous phase was added, and the mixtures were sonicated using a rod sonicator (Q500 of QSonica^®^, Newtown, CT, USA) in batch mode for 25 min at 30 s intervals every minute, in an ice bath. Each formulation containing bioactive(s) was centrifuged at 11,180× *g* for 15 min to eliminate particles released by the titanium rod sonicator.

Nanostructured lipid system formulations containing 10 mg/mL metformin or 9 mg/mL lycopene (or both when in combination) were prepared, allowing the formulations to reach the doses of 50 mg/kg metformin and 45 mg/kg lycopene, respectively (or both when in combination).

### 4.2. Animal Experimental Design

C57BL-6J male mice (4-week-old; weight 21 ± 0.17 g) were obtained from ANILAB (Animais de Laboratório, Criação e Comércio LTDA, Paulínia, Brazil). During the experimental period, animals were housed into polypropylene cages (two animals per cage) under controlled conditions of temperature (23 ± 1 °C) and humidity (55% ± 5%) with a 12/12 h light/dark cycle. During the acclimation period, mice received a standard chow diet (Presence Nutrição Animal, Paulínia, São Paulo, Brazil) and water ad libitum. After 2 weeks of acclimation, mice were fed with control diet (C; 3.85 kcal/g; 4% lipids) or high-fat diet (HFD; 5.40 kcal/g; 35% lipids) (Pragsoluções Biociências Serviços e Comércio Ltd.a, Jaú, São Paulo, Brazil) (Appendix A). HFD enabled the development of obesity.

Mice were divided into six groups (n = 12 per group; 72 animals in total): C group (non-obese, C diet-fed mice), H group (obese, HFD-fed mice), H-V group (HFD-fed mice treated with vehicle), H-M group (HFD-fed mice treated with 50 mg/kg metformin), H-L group (HFD-fed mice treated with 45 mg/kg lycopene), and H-ML group (HFD-fed mice treated with 50 mg/kg metformin + 45 mg/kg lycopene). 

Stratified randomization was used to allocate mice to experimental groups. Additionally, the HFD groups were composed of mice that would standardize the median body weight across the groups. For experimental groups having obese animals, inclusion criteria were based on mice having body weight values around 30 g before the beginning of treatments. As exclusion criteria, the low body weight gain values of mice fed the HFD were used.

All treatments were performed by oral administration (gavage). Doses of metformin (50 mg/kg) [41] and lycopene (45 mg/kg) [21,42] were chosen based on our previous studies. Tomato extract powder (*Lycopersicon esculentum* Mill., 10.13% lycopene, Florien Fitoativos, Piracicaba, São Paulo, Brazil) and metformin (99.56% metformin hydrochloride, Abhilash Chemicals and Pharmaceuticals, Madurai, Tamilnadu, India) were incorporated in a nanostructured lipid system.

The study duration was 17 weeks. The treatments were performed during the last 8 weeks and administered by daily gavage (5 µL/g animal), between 08:00–09:00, from the 9th to the 17th weeks. Mice from H-V group received vehicle by daily gavage (5 µL/g animal), which consisted of the nanostructured lipid system without metformin or lycopene. Mice from C and H groups received water by gavage (to mimic the treatments). Body weight and food intake were assessed weekly. Energy intake was calculated by multiplying food intake (g) by the energy values (kcal) provided for the diets.

Oral glucose tolerance test (OGTT) and insulin tolerance test (ITT) were performed at weeks 15 and 16, respectively (corresponding to 6 and 7 weeks of treatment), according to Talpo et al. [41]. Briefly, for the OGTT, mice were fasted for 12 h, and the OGTT was performed at 11:00 a.m. after gavage with 1 g/kg glucose. Glycemia levels were monitored before (0 min) and after (15, 30, 60, 90, and 120 min) the glucose challenge. A drop of blood was obtained from the mouse’s tail to determine the glycemia levels measured by glucometer (Abbott Diabetes Care Ltd., São Paulo, Brazil). For the ITT, mice were fasted for 2 h, and the ITT was performed at 01:00 p.m. with an intraperitoneal injection of insulin (0.4 UI/kg). Glycemia levels were monitored before (0 min) and after (15, 30, 45, and 60 min) the insulin injection.

After 17 weeks (corresponding to 8 weeks of treatment), mice were subjected to 6 h fasting, following which they were anesthetized (16 mg/kg xylazine and 90 mg/kg ketamine), and blood samples were collected in heparinized tubes and centrifuged immediately (700× *g* for 10 min at 25 °C) to obtain plasma samples. Epididymal white adipose tissue (eWAT), retroperitoneal white adipose tissue (rWAT), interscapular brown adipose tissue (iBAT), gastrocnemius skeletal muscles, heart, liver, and kidneys were removed, weighed, snap-frozen in liquid nitrogen, and immediately frozen (–80 °C).

All experimental procedures were previously approved by the Committee for Ethics in Animal Experimentation at the School of Pharmaceutical Sciences, Unesp (CEUA/FCF/CAr n° 43/2018).

### 4.3. Plasma Biochemical Analysis

Plasma levels of glucose, triglycerides, total-cholesterol, high-density lipoprotein-cholesterol (HDL-cholesterol), alanine aminotransferase (ALT), alkaline phosphatase (ALP), albumin, and creatinine were measured using commercial kits (Labtest Diagnostica SA, Lagoa Santa, Brazil).

### 4.4. Biomarkers of Glycoxidative Stress 

Levels of fluorescent AGEs in the plasma and liver were assessed according to Zilin et al. [43] with modifications, and the fluorescent AGEs in the kidneys were measured according to Pokupec et al. [44]. Briefly, 0.12 M trichloroacetic acid, 0.1 M sodium hydroxide, and 1.2 M chloroform (plasma) or 2.4 M chloroform (liver) were added to plasma and liver homogenates. An amount of 0.1 M sodium hydroxide was added to kidney homogenates. Tubes containing these samples were shaken vigorously, maintained at 10 ± 2 °C for 30 min, and centrifuged at 10,000× *g* for 10 min at 10 °C. Fluorescence intensities relative to AGEs in the supernatants were measured spectrofluorometrically, with excitation and emission wavelengths of 370 and 440 nm, respectively, by using a microplate multi-mode reader with split set at 16 nm (Synergy H1TM, BioTek Instruments Inc, Winooski, VT, USA). The results were expressed as arbitrary units (AU) of fluorescence per milligram of protein.

Lipid peroxidation products, including malondialdehyde, were measured in deproteinized tissue samples (plasma, liver, and kidneys) using the thiobarbituric acid (TBA) assay according to Kohn and Liversedge [45]. TBA reacts mainly with malondialdehyde, which generates products named thiobarbituric acid reactive substances (TBARS), whose levels were measured spectrophotometrically (535 nm) in liver and kidneys, or spectrofluorometrically (excitation and emission wavelengths of 510 nm and 553 nm, respectively) in plasma. The results were expressed in terms of µmol/L (plasma) and nmol/g tissue (liver and kidneys).

### 4.5. Analysis of the Antioxidant Defenses

The activities of the antioxidant enzymes superoxide dismutase (SOD), catalase (CAT), and glutathione peroxidase (GSH-Px) were measured in the liver and kidneys, and the activity of paraoxonase-1 (PON-1) was measured in the plasma, according to standardized methods.

Sample preparations for liver and kidneys were performed according to the following: liver and kidneys (0.1 g) were homogenized in 1 mL sodium phosphate buffer (10 mmol/L, pH 7.4) at 4 °C. Homogenates were centrifuged at 10,000 × *g* for 10 min at 4 °C, and the supernatants were used for the analysis of the activities of SOD, CAT and GSH-Px.

The SOD activity was evaluated by monitoring the inhibition of the nitroblue tetrazolium chloride (NBT) reduction by the superoxide anion radical generated by the xanthine oxidation reaction in the presence of xanthine oxidase. SOD enzymes present in the sample catalyzes the dismutation of superoxide anion radicals and inhibits the NBT reduction, which is monitored spectrophotometrically at 550 nm [46]. The results were expressed in terms of U/mg protein. One SOD unit is defined as the enzyme amount required to inhibit the rate of NBT reduction by 50%.

The CAT activity was measured by monitoring the consumption of hydrogen peroxide (H_2_O_2_) spectrophotometrically at 230 nm [47]. The results were expressed in terms of μmol of H_2_O_2_ consumed/min/mg protein.

The GSH-Px activity was determined by monitoring the oxidation of nicotinamide adenine dinucleotide phosphate, reduced form (NADPH) to NADP+. In the presence of H_2_O_2_, GSH-Px catalyzes the oxidation of the metabolite glutathione (GSH). The glutathione in its oxidized form (GSSG) is reduced to GSH by the reaction catalyzed by glutathione reductase, with the concomitant oxidation of NADPH to NADP+, which is monitored spectrophotometrically at 340 nm [48]. The results were expressed in terms of μmol of NADPH oxidized/min/mg protein.

The activity of PON-1 in plasma samples was measured according to Costa et al. [49], with modifications made by Assis et al. [42]. Plasma PON-1 activity was measured by the hydrolysis of paraoxon and release of *p*-nitrophenol, which was monitored at 405 nm. The results were expressed in terms of U/mg HDL-cholesterol (unit = μmoL paraoxon hydrolyzed/min).

Protein levels (plasma or tissue supernatants) were determined according to Lowry et al. [50] for the correction of the results related to SOD, CAT, GSH-Px, and the fluorescent AGEs.

### 4.6. Statistical Analysis

Data are expressed in terms of mean ± standard error of mean (SEM). One-way analysis of variance followed by the Student-Newman-Keuls test was used to compare the inter-group differences. Paired Student’s *t*-test was used to compare the intra-group changes in the body weight values, relative to week 0. Data were considered statistically different at *p* < 0.05. Statistical analyses were performed using GraphPad Prism 6.01 (GraphPad Software, San Diego, CA, USA).

## 5. Conclusions

Combination therapy of current medicines with natural antioxidants might be an interesting strategy to contrast the events related to metabolic memory and the long-term complications of metabolic diseases. Therefore, the present study showed the beneficial effects of metformin combined with lycopene on obese mice, not only improving insulin resistance and glucose tolerance, but also decreasing glycoxidative stress and increasing endogenous antioxidant defenses. The significant improvement in the activity of PON-1 sheds a light to the promising therapeutic effectiveness of metformin + lycopene for the management of metabolic complications resulting from oxidative stress, especially cardiovascular diseases.

## Figures and Tables

**Figure 1 molecules-27-08503-f001:**
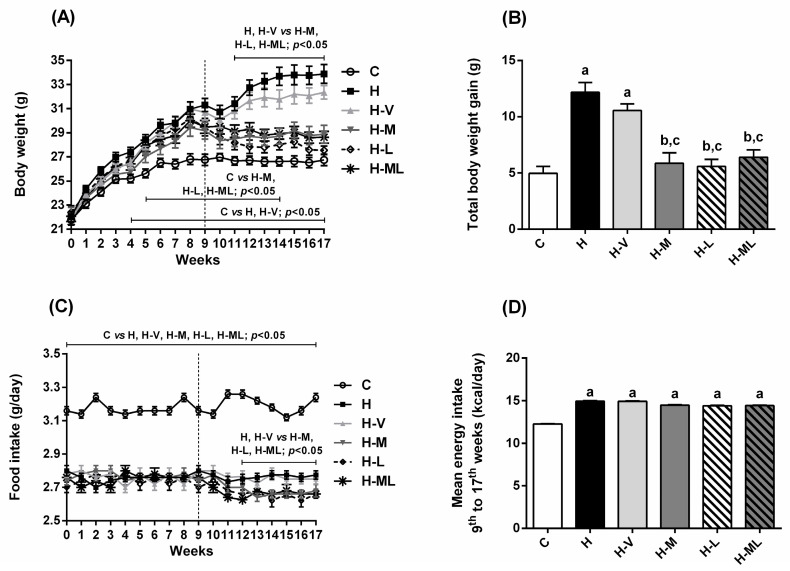
Body weight and food and energy intakes of HFD-fed mice treated for 8 weeks with metformin, alone or in combination with lycopene. Body weight (**A**), total body weight gain (**B**), food intake (**C**), and mean energy intake (**D**). Values are expressed as mean ± SEM, n = 12. C: mice fed control diet; H: mice fed HFD; H-V: mice fed HFD and treated with vehicle; H-M: mice fed HFD and treated with 50 mg/kg metformin; H-L: mice fed HFD and treated with 45 mg/kg lycopene; H-ML: mice fed HFD and treated with 50 mg/kg metformin + 45 mg/kg lycopene. Differences between groups were considered significant at *p* < 0.05. a, differences with C; b, differences with H; c, differences with H-V.

**Figure 2 molecules-27-08503-f002:**
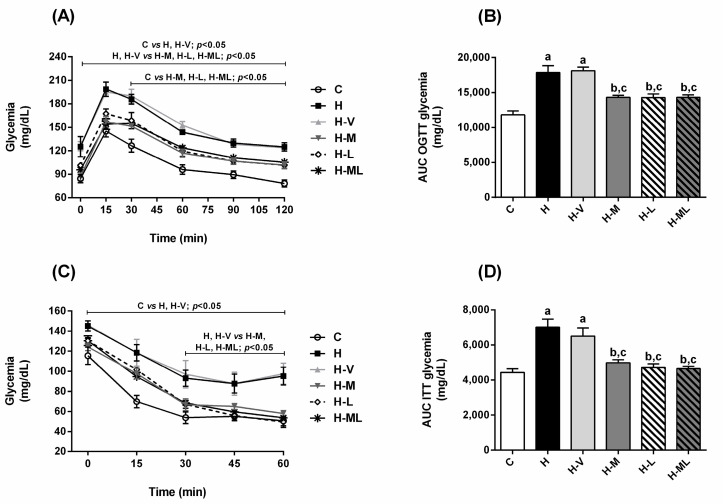
Glucose tolerance and insulin sensitivity of HFD-fed mice treated for 8 weeks with metformin, alone or in combination with lycopene. Oral glucose tolerance test (OGTT) (**A**), AUC of OGTT (**B**), insulin tolerance test (ITT) (**C**), and AUC of ITT (**D**). Values are expressed as mean ± SEM, n = 12. C: mice fed control diet; H: mice fed HFD; H-V: mice fed HFD and treated with vehicle; H-M: mice fed HFD and treated with 50 mg/kg metformin; H-L: mice fed HFD and treated with 45 mg/kg lycopene; H-ML: mice fed HFD and treated with 50 mg/kg metformin + 45 mg/kg lycopene. AUC: area under the curve. Differences between groups were considered significant at *p* < 0.05. a, differences with C; b, differences with H; c, differences with H-V.

**Figure 3 molecules-27-08503-f003:**
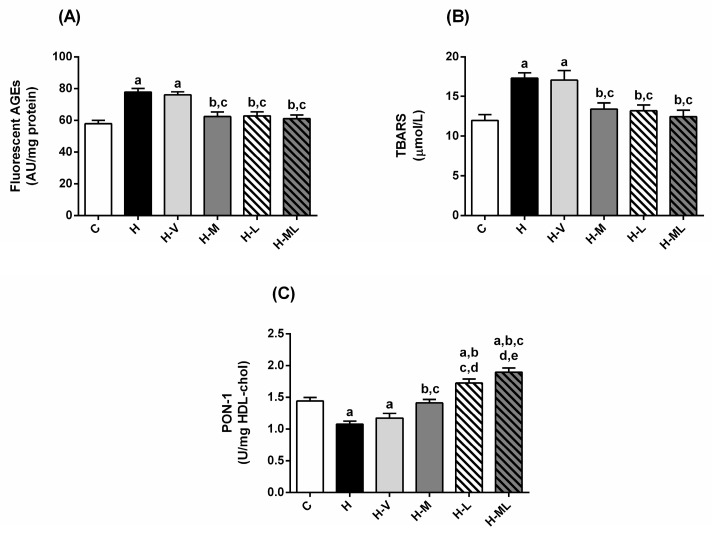
Biomarkers of glycoxidative damage and antioxidant defenses in plasma of HFD-fed mice treated for 8 weeks with metformin, alone or in combination with lycopene. Fluorescent AGEs (**A**), TBARS (**B**), and PON-1 activity (**C**). Values are expressed as mean ± SEM, n = 12. C: mice fed control diet; H: mice fed HFD; H-V: mice fed HFD and treated with vehicle; H-M: mice fed HFD and treated with 50 mg/kg metformin; H-L: mice fed HFD and treated with 45 mg/kg lycopene; H-ML: mice fed HFD and treated with 50 mg/kg metformin + 45 mg/kg lycopene. Differences between groups were considered significant at *p* < 0.05. a, differences with C; b, differences with H; c, differences with H-V; d, differences with H-M; e, differences with H-L.

**Figure 4 molecules-27-08503-f004:**
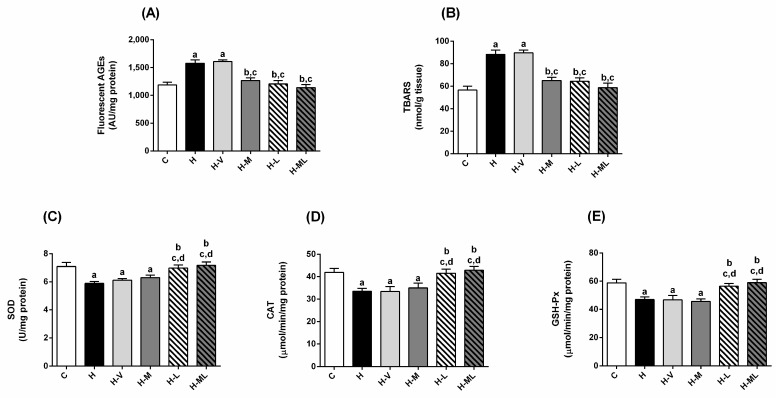
Biomarkers of glycoxidative damage and antioxidant defenses in liver of HFD-fed mice treated for 8 weeks with metformin, alone or in combination with lycopene. Fluorescent AGEs (**A**), TBARS (**B**), and activities of SOD (**C**), CAT (**D**) and GSH-Px (**E**). Values are expressed as mean ± SEM, n = 12. C: mice fed control diet; H: mice fed HFD; H-V: mice fed HFD and treated with vehicle; H-M: mice fed HFD and treated with 50 mg/kg metformin; H-L: mice fed HFD and treated with 45 mg/kg lycopene; H-ML: mice fed HFD and treated with 50 mg/kg metformin + 45 mg/kg lycopene. Differences between groups were considered significant at *p* < 0.05. a, differences with C; b, differences with H; c, differences with H-V; d, differences with H-M.

**Figure 5 molecules-27-08503-f005:**
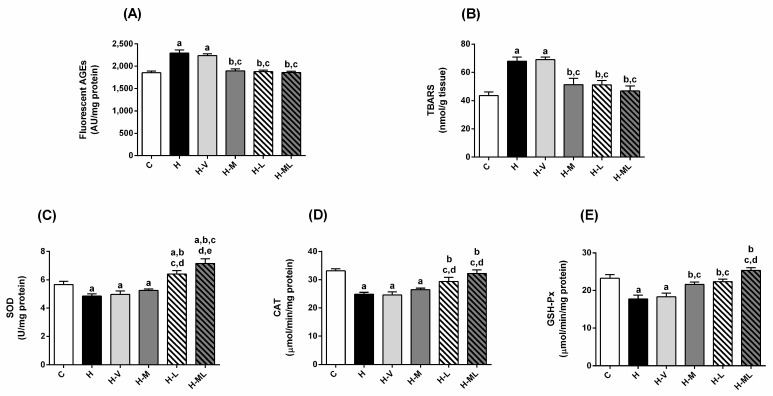
Biomarkers of glycoxidative damage and antioxidant defenses in kidney of HFD-fed mice treated for 8 weeks with metformin, alone or in combination with lycopene. Fluorescent AGEs (**A**), TBARS (**B**), and activities of SOD (**C**), CAT (**D**) and GSH-Px (**E**). Values are expressed as mean ± SEM, n = 12. C: mice fed control diet; H: mice fed HFD; H-V: mice fed HFD and treated with vehicle; H-M: mice fed HFD and treated with 50 mg/kg metformin; H-L: mice fed HFD and treated with 45 mg/kg lycopene; H-ML: mice fed HFD and treated with 50 mg/kg metformin + 45 mg/kg lycopene. Differences between groups were considered significant at *p* < 0.05. a, differences with C; b, differences with H; c, differences with H-V; d, differences with H-M; e, differences with H-L.

**Table 1 molecules-27-08503-t001:** Body weight (g) at weeks 0, 9, and 17 and body weight gain (g) (from 0 to 9 weeks, and from 9 to 17 weeks) of HFD-fed mice treated for 8 weeks with metformin, alone or in combination with lycopene.

Groups	C	H	H-V	H-M	H-L	H-ML
Body weight (g) at week 0 (beginning of experiment)	21.77 ± 0.421	22.04 ± 0.412	21.74 ± 0.275	22.25 ± 0.563	22.39 ± 0.559	21.65 ± 0.413
Body weight (g) at week 9 (beginning of treatments)	26.90 ± 0.412 (#)	31.30 ± 0.546 (#,a)	30.02 ± 0.712 (#,a)	30.01 ± 0.719 (#,a)	30.11 ± 0.697 (#,a)	30.27 ± 0.413 (#,a)
Body weight gain (g) (week 0 → week 9)	5.12 ± 0.525	9.26 ± 0.398	8.71 ± 0.638	7.76 ± 0.635	7.71 ± 0.440	8.61 ± 0.523
Body weight (g) at week 17 (end of treatments)	26.74 ± 0.477 (#)	33.88 ± 0.744 (#,a)	32.36 ± 0.633 (#,a)	28.87 ± 0.749 (#,b,c)	27.55 ± 0.351 (#,b,c)	28.62 ± 0.495 (#,b,c)
Body weight gain (g) (week 9 → week 17)	−0.16 ± 0.361	2.73 ± 0.387	2.49 ± 0.636	−1.15 ± 0.482	−2.65 ± 0.729	−1.63 ± 0.480

Values are expressed as mean ± standard error of mean (SEM), n = 12. C: mice fed control diet; H: mice fed HFD; H-V: mice fed HFD and treated with vehicle; H-M: mice fed HFD and treated with 50 mg/kg metformin; H-L: mice fed HFD and treated with 45 mg/kg lycopene; H-ML: mice fed HFD and treated with 50 mg/kg metformin + 45 mg/kg lycopene. Differences between groups were considered significant at *p* < 0.05. a, differences with C; b, differences with H; c, differences with H-V. Differences in the same group relative to week 0 were analyzed using the paired Student’s *t*-test (*p* < 0.05): #, differences with week 0.

**Table 2 molecules-27-08503-t002:** Weights of tissues of HFD-fed mice treated for 8 weeks with metformin, alone or in combination with lycopene.

Groups	C	H	H-V	H-M	H-L	H-ML
Epididymal WAT (mg/mm tibia)	26.74 ± 1.474	73.04 ± 5.432 (a)	53.74 ± 3.980 (a,b)	37.20 ± 3.262 (b,c)	36.44 ± 2.89 (b,c)	37.97 ± 3.243 (b,c)
Retroperitoneal WAT (mg/mm tibia)	11.30 ± 0.798	30.62 ± 1.797 (a)	21.52 ± 1.775 (a,b)	14.02 ± 1.257 (b,c)	12.78 ± 0.853 (b,c)	15.13 ± 1.213 (b,c)
Interscapular BAT (mg/mm tibia)	3.72 ± 0.144	3.97 ± 0.311	3.56 ± 0.240	3.17 ± 0.204	3.31 ± 0.251	3.34 ± 0.218
Liver (mg/mm tibia)	57.95 ± 0.872	67.16 ± 1.573 (a)	62.87 ± 2.126 (a)	57.37 ± 1.805 (b,c)	57.08 ± 1380 (b,c)	57.64 ± 1.492 (b,c)
Kidney (mg/mm tibia)	8.23 ± 0.537	10.98 ± 0.656 (a)	10.10 ± 0.766 (a)	8.78 ± 0.579 (b,c)	8.88 ± 0.748 (b,c)	8.79 ± 0.395 (b,c)
Heart (mg/mm tibia)	7.08 ± 0.206	7.22 ± 0.108	7.11 ± 0.172	7.13 ± 0.160	7.04 ± 0.140	7.20 ± 0.180
Gastrocnemius muscle (mg/mm tibia)	8.08 ± 0.079	8.31 ± 0.174	8.11 ± 0.140	8.12 ± 0.103	8.16 ± 0.103	8.31 ± 0.152

Values are expressed as mean ± SEM, n = 12. C: mice fed control diet; H: mice fed HFD; H-V: mice fed HFD and treated with vehicle; H-M: mice fed HFD and treated with 50 mg/kg metformin; H-L: mice fed HFD and treated with 45 mg/kg lycopene; H-ML: mice fed HFD and treated with 50 mg/kg metformin + 45 mg/kg lycopene. Differences between groups were considered significant at *p* < 0.05. a, differences with C; b, differences with H; c, differences with H-V.

**Table 3 molecules-27-08503-t003:** Levels of biochemical parameters in plasma of HFD-fed mice treated for 8 weeks with metformin, alone or in combination with lycopene.

Groups	C	H	H-V	H-M	H-L	H-ML
Triglycerides (mg/dL)	52.1 ± 20.6	41.2 ± 2.08 (a)	40.0 ± 2.37 (a)	38.9 ± 3.65 (a)	37.8 ± 2.96 (a)	38.8 ± 2.25 (a)
Total-cholesterol (mg/dL)	98.3 ± 3.25	137.3 ± 3.28 (a)	127.8 ± 4.88 (a)	107.2 ± 4.33 (b,c)	108.3 ± 3.13 (b,c)	109.0 ± 2.97 (b,c)
HDL-cholesterol (mg/dL)	79.3 ± 2.52	108.6 ± 2.43 (a)	100.6 ± 2.33 (a)	87.9 ± 4.00 (b,c)	84.7 ± 4.21 (b,c)	80.7 ± 3.84 (b,c)
ALT (UI/L)	21.9 ± 1.27	32.9 ± 2.47 (a)	31.5 ± 1.26 (a)	20.5 ± 1.22 (b,c)	20.3 ± 1.74 (b,c)	20.6 ± 1.53 (b,c)
ALP (UI/L)	41.7 ± 1.96	54.6 ± 3.49 (a)	52.3 ± 1.74 (a)	42.4 ± 2.60 (b,c)	42.0 ± 2.75 (b,c)	44.5 ± 1.22 (b,c)
Albumin (g/dL)	2.13 ± 0.021	2.12 ± 0.028	2.14 ± 0.025	2.09 ± 0.043	2.08 ± 0.028	2.12 ± 0.027
Creatinine (mg/dL)	0.16 ± 0.003	0.15 ± 0.002	0.16 ± 0.004	0.15 ± 0.003	0.16 ± 0.004	0.15 ± 0.003

Values are expressed as mean ± SEM, n = 12. C: mice fed control diet; H: mice fed HFD; H-V: mice fed HFD and treated with vehicle; H-M: mice fed HFD and treated with 50 mg/kg metformin; H-L: mice fed HFD and treated with 45 mg/kg lycopene; H-ML: mice fed HFD and treated with 50 mg/kg metformin + 45 mg/kg lycopene. Differences between groups were considered significant at *p* < 0.05. a, differences with C; b, differences with H; c, differences with H-V.

## Data Availability

The datasets generated during and/or analyzed during the current study are available from the corresponding author on reasonable request.

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
