# Peer review of "Combined Effects of Lycopene and Metformin on Decreasing Oxidative Stress by Triggering Endogenous Antioxidant Defenses in Diet-Induced Obese Mice"

_molecules, 2022, doi:10.3390/molecules27238503_

Round 1

Reviewer 1 Report

Dear Author,

Original manuscript entitled " Lycopene aggregates antioxidant properties to metformin treatment by triggering endogenous antioxidant defenses in obese mice” targets a topic that fits well within the journal scope and that is potentially very interesting to the journal readers. This study aimed to investigate the effect of combined treatment with metformin and lycopene on insulin resistance, glucose intolerance, and endogenous antioxidant defenses in obese/insulin resistant mice. While I appreciated reading this manuscript few issues need to be addressed before the manuscript meets the journal standards:

Please, rephrase the title – I think the word ‘aggregates’ is not the best choice

The abstract is too long, it should be a maximum of 200 words.

l.42-46 and 48-51, 56 - 59 and other– the sentences are too long and need rephrasing

l.73 Mice fed the HFD for 17 weeks (H group)  - what is HFD or C diet? – abbreviation is not explained, also in Table 1 – HFD abbreviation is not explained

l. 80 what is eWAT and rWAT? – it’s not explained where it first appears

In general: abbreviations should be defined the first time they appear in each of three sections: the abstract; the main text; the first figure or table – The abbreviations are explained later in the text

Please, argue, why you selected lycopene, and not a different carotenoid.

What may be the mode of action of the effect of lycopene on glycation, in what way may it enhance antioxidant enzymes (transcriptional, post-transcriptional)?

In discussion please, mention if there are any data available on this topic in humans.

I am looking towards receiving an improved version of this manuscript that addresses all these issues .

Best regards

Reviewer 2 Report

The authors present a study that explores the combination of an oral antidiabetic drug of the biguanide type and a carotenoid with antioxidant properties, in the treatment of some metabolic dysfunctions. Lycopene and metformin are supplied in a lipid nanostructured system preparation. The application aims to give a more natural response to the treatment of the aforementioned dysfunctions. Improvement in insulin resistance, glucose intolerance, and endogenous antioxidant defenses were sought in obese/insulin-resistant mice. Metformin plus lycopene treatment reduced body weight gain and fat deposition; with improved insulin sensitivity and glucose tolerance; decreasing the levels of biomarkers of glucoxidative stress. Although the study is at the preclinical level, it is a promising approach to combat the metabolic dysfunctions that occur as a result of glucoxidative stress. The Work is novel and its publication is recommended once the observations expressed here have been solved.

1.- It is necessary to detail more why it should be supplied in this preparation, and what advantages the nanostructured lipid system has in the preparation (Introduction) and its possible contribution to the results observed (what is mentioned in lines 301 to 306 is very poor and does not contribute anything substantial to the understanding). Which should be reflected in the final manuscript.

2.- lines 265 to 269, the authors argue that “In agreement with this, our findings showed that 265 lycopene + metformin significantly improved the glucose tolerance and insulin sensitivity 266 in obese mice; such effects were virtually the same as those of metformin or lycopene alone”. If the last part is true, then there is no synergistic delivery effect of metformin and lycopene and this study is pointless.

3.- Lines 276 to 278 is very poor in discussion, go deeper and see it reflected in the document

4.- Lines 330 to 333. The preparation of the lipid nanostructured system and its stability over time must be further detailed.

Reviewer 3 Report

Add importance of the study in start of the abstract

Give range of obesity or give weight range of obese mice in line 16.

Add statistical analysis and study design in the abstract

Add quantitative results in the abstract so that visibility and understanding of your article could be better.

 Add conclusive line in the end of abstract.

Please be specific in introduction section no need to discuss other diseases in it just focus on your target diseases.

T2DM is more as compared to obesity, try to add some more data and its prevalence in the introduction,

Add rationale and reasoning of the study in the last paragraph of introduction.

Statistical analysis is not clear can you please elaborate LSD in your results section.

Discussion needs proper comparison with previous studies and give justification as well.

Please mention the dose preparation in methodology and give their levels.

Please explain about the remaining analysis why you have not focused on weight gain and loss during the study or pre and post weight of mice because that is the important parameter of obesity as well as diabetes, so please mention their results as well.

Explain the conditions of animal room.

Give distribution and inclusion exclusion criteria of your study.

Grammatically this paper needs your serious attention please thoroughly read the article and improve it.

Round 2

Reviewer 1 Report

Dear Authors, 

Thank you for your responses to reviewer's comments. The manuscript is well revised. 

Best regards

Author Response

Dear Reviewer,

We greatly appreciate all the comments previously made by the Reviewer, which produced substantial improvements in the quality of the manuscript.

In addition, English revision was performed in all sections of the manuscript.

Reviewer 3 Report

Thank you for your Responses.

I am agreed with your answers but one thing that is very important Author should understand that little methodology in abstract is very important. So please add one or two lines of methodology in the abstract section. 

Overall article is well revised.

Author Response

Dear Reviewer,

We are fully in agreement with the reviewer comment.

We have included new information about Methodology in the “Abstract”.

Please see the changes that are highlighted in "yellow" in the new version of the Abstract.

In addition, English revision was performed in all sections of the manuscript.